# Learning reliably under adversarial attacks, distribution shifts and strategic behavior

**Maria-Florina Balcan**
Carnegie Mellon University
ninamf@cs.cmu.edu

**Dravyansh Sharma**
Toyota Technological Institute at Chicago
dravy@ttic.edu

## Abstract

The impressive strengths of generative AI brings along with it a host of new challenges since the tool is equally available to good and bad actors. For example, malicious agents can use it to create a large number of unreliable online reviews with very low effort, enabling more aggressive targeted attacks on recommendation systems. Therefore, learning models that depend on this data must incorporate strong reliability guarantees about their predictions to be at all useful. A recent line of work initiates this important direction by formalizing these reliability guarantees, along with tight upper and lower bounds on when they may be achieved. Initiated in the context of poisoning attacks on the training data, follow-up works have successfully shown how similar guarantees may be given for test-time adversarial attacks, distribution shifts and strategic manipulations. We discuss several future directions for this line of research.

## 1 Introduction

Machine learning has become pervasive in decision-making, often in settings where reliability is paramount, including medical diagnosis, content moderation, autonomous driving, and financial forecasting. In these contexts, standard PAC learning guarantees are insufficient. It is not enough that a hypothesis has low error on average; each individual prediction may need a certificate of correctness saying that the predictions are reliable even under sophisticated and targeted adversarial attacks or other challenges that may impact the correctness of predictions. Recent work [Balcan et al., 2022, 2023b] initiates a formal framework for providing such guarantees against both training-time and test-time attacks, as well as understanding when it is possible to give such guarantees. Blum et al. [2025] further show how to make these reliable guarantees more explainable by generating short proofs of correctness, enhancing the trustworthiness of the models.

The advent of large language models and generative AI makes the need for reliability even more urgent. Unlike traditional classifiers that output a single label, LLMs generate long, nuanced responses that can directly influence human decision-making in high-stakes domains. Even a small fraction of incorrect or misleading outputs can have outsized consequences. Moreover, LLMs are vulnerable to both training-time contamination (e.g., data poisoning or backdoored prompts) and test-time adversaries (e.g., prompt injection attacks), echoing the challenges addressed in robustly-reliable learning. Extending per-prediction guarantees, abstention mechanisms, and explanatory certificates to the generative setting could provide a foundation for certifiably trustworthy AI systems that not only produce fluent outputs, but also signal when they can be relied upon—and crucially, when they should not.

Over the past few years, a sequence of works have developed the formal framework of *robustly-reliable learning*. This line of work represents a conceptual shift from "aggregate accuracy" to "per-instance guarantees", unifying insights from standard PAC learning, selective classification, ad-

39th Conference on Neural Information Processing Systems (NeurIPS 2025) Workshop: Reliable ML from Unreliable Data.

versarial robustness, explainability, and complexity measures like disagreement coefficient introduced in the context of active learning [Balcan and Urner, 2014]. In this perspective, we summarize the foundational ideas, highlight some follow-up extensions, and propose a forward-looking research agenda to significantly expand on these initial works.

## 2 Robustly-reliable learning

Rivest and Sloan [1988] introduced an extension of standard PAC learning to *reliable and useful* learning, where the learners are allowed to abstain ("say $\perp$", the reject option was introduced by Chow [1970]) rather than risk an incorrect prediction. Their motivation came from safety-critical tasks, where returning no decision is preferable to returning the wrong one. The key insight was to move beyond minimizing expected error toward frameworks that explicitly trade off coverage or usefulness (the fraction of points labeled) and confidence or reliability (the correctness of non-abstained predictions). This generalizes the related notion of learning with one-sided error [Natarajan, 1987, Kivinen, 1995, Bshouty and Burroughs, 2005], where the learner is only allowed to make errors of only one kind (abstains can be replaced by a default to the label class for which error is permitted), capturing scenarios where false positives might be much more harmful than false negatives (or vice versa). Reliable learning has also been called selective classification [El-Yaniv and Wiener, 2012].

The modern notion of robustly-reliable learning builds directly on this foundation. It retains abstention as a central mechanism, but strengthens the guarantees by requiring correctness to hold robustly under adversarial corruptions and perturbations. Balcan et al. [2022, 2023b] achieve this by designing learners that output for every point not only a prediction, but an accompanying "certification level" which can depend on the specific type of reliability guarantee. For example, a reliability guarantee against a poisoning attack would be a "certification level" $\eta$ which implies that the learner's prediction for that point is guaranteed to be correct, provided the adversary corrupted (poisoned) at most an $\eta$ fraction of the training data (including arbitrary insertions, deletions or substitutions). Note that the learner provides a per-point certificate that applies even if the attack is *instance-targeted*, that is, the adversary uses their entire attack budget just to cause an error at the given point (as opposed to maximizing average error). Thus, the reliability guarantee provides simultaneous security against different "highly targeted" attacks for all possible targets. In this way, robustly-reliable learning extends Rivest and Sloan's notion of reliability to meet the demands of modern robustness theory, extending selective classification into settings with adversaries, distribution shift, and strategic agents.

We emphasize that these guarantees are fundamentally different from those studied in the theoretical literature on adversarial robustness [e.g., Yin et al., 2019, Attias et al., 2019, Montasser et al., 2019, 2020, 2021, Goldwasser et al., 2020], which mainly focus on test-time attacks and typically study average error instead of per-point guarantees against targeted attacks. Even classical works on malicious noise models (which capture training-time attacks) have focused exclusively on average error [Kearns and Li, 1993, Bshouty et al., 2002, Klivans et al., 2009, Awasthi et al., 2017]. In contrast, instance targeted poisoning attacks are more challenging as the adversary can use the entire corruption budget to target specific instances, making them more suitable for modern applications like fake review detectors where a malicious adversary is more likely to post fabricated reviews praising their own product or attacking a specific competitor. A related line of work [Levine and Feizi, 2021, Gao et al., 2021] provides per-point "stability" guarantees that the predictions are unchanged under the attacks, but it does not provide any correctness guarantees. The same limitation applies to the approach of Cohen et al. [2019] developed in the practical context of deep learning.

We will now summarize and illustrate the formal setup for robustly-reliable learning for test-time attacks. See Appendix A for the corresponding definition for poisoning attacks.

Let $\mathcal{X}$ denote the instance space and $\mathcal{Y} = \{0, 1\}$ be the label space. Let $\mathcal{H}$ be a hypothesis class. The learner $\mathcal{L}$ is given access to a labeled sample $S = \{(x_i, y_i)\}_{i=1}^m$ drawn from a distribution $\mathcal{D}$ over $\mathcal{X} \times \mathcal{Y}$ and learns a concept $h_S^{\mathcal{L}} : \mathcal{X} \rightarrow \mathcal{Y}$. In the realizable setting, we assume we have a hypothesis (concept) class $\mathcal{H}$ and target concept $h^* \in \mathcal{H}$ such that the *true label* of any $x \in \mathcal{X}$ is given by $h^*(x)$. In particular, $S = \{(x_i, h^*(x_i))\}_{i=1}^m$ in this setting. Given the 0-1 loss function $\ell : \mathcal{H} \times \mathcal{X} \rightarrow \{0, 1\}$, define $\text{err}_S(h, \ell) = \frac{1}{m} \sum_{(x,y) \in S} \ell(h, x)$. Use $\mathbb{I}[\cdot]$ to denote the indicator function that takes values in $\{0, 1\}$. We also define $B_{\mathcal{D}}^{\mathcal{H}}(h^*, r) = \{h \in \mathcal{H} \mid \Pr_{\mathcal{D}}[h(x) \neq h^*(x)] \leq r\}$ as the set of hypotheses in $\mathcal{H}$ that disagree with $h^*$ with probability at most $r$. During test-time, the learner makes a prediction on a test-point $z \in \mathcal{X}$. We consider adversarial attacks with perturbation function $\mathcal{U} : \mathcal{X} \rightarrow 2^{\mathcal{X}}$ that

can perturb a test point $x$ to an arbitrary point $z$ from the perturbation set $\mathcal{U}(x) \subseteq \mathcal{X}$ [Montasser et al., 2019]. A case of particular interest is where the perturbation set corresponds to a ball in a metric space. That is, $\mathcal{U}_{\mathcal{M}} = \{u_\eta : \mathcal{X} \to 2^{\mathcal{X}} \mid u_\eta(x) = \mathbf{B}_{\mathcal{M}}(x, \eta)\}$, induced by the metric $\mathcal{M} = (\mathcal{X}, d)$ defined over the instance space. We use $\mathbf{B}^o_{\mathcal{M}}(x, r) = \{x' \in \mathcal{X} \mid d(x, x') < r\}$ to denote the open ball of radius $r$ centered at $x$. We assume that the adversary has access to the learned concept $h^{\mathcal{L}}_S$ as well as the test point $x$, and can perturb this data point to any $z \in \mathcal{U}(x)$ and then provide this perturbed data point to the learner at test-time. We also assume that $x \in \mathcal{U}(x)$ for all $x \in \mathcal{X}$.

In the applied and theoretical literature, various definitions of adversarial success have been explored, each dependent on the interpretation of robustness. For example, the adversary may perturb a test point $x$ to any point $z$ in the perturbation set $\mathcal{U}(x)$, but is not allowed to change its true label $h^*(x)$. The adversary succeeds if the learned classifier $h$ is incorrect on $z$. This is also called the *constrained adversary loss* [Szegedy et al., 2014, Balcan et al., 2023a].

$$\ell^{h^*}(h, x) = \sup_{z \in \mathcal{U}(x),\ h^*(z) = h^*(x)} \mathbb{I}[h(z) \neq h^*(z)].$$

For a fixed perturbation $z \in \mathcal{U}(x)$, define $\ell^{h^*}(h, x, z) = \mathbb{I}[h(z) \neq h^*(z) \wedge h^*(z) = h^*(x)]$. Balcan et al. [2023b] study robustness w.r.t. this and several other robust and strategic losses in the literature.

At test-time, given a test-point $z \in \mathcal{X}$, we would like to make a prediction at $z$ with a reliability guarantee. We consider this type of learner, a *robustly-reliable* learner, defined formally as follows.

**Definition 1 (Robustly-reliable learner w.r.t. $\mathcal{M}$-ball attacks)** *A learner $\mathcal{L}$ is robustly-reliable w.r.t. $\mathcal{M}$-ball attacks for hypothesis space $\mathcal{H}$ and robust loss function $\ell$ if, **for any target** concept $h^* \in \mathcal{H}$, given $S$ labeled by $h^*$, the learner outputs two functions $h^{\mathcal{L}}_S : \mathcal{X} \to \mathcal{Y}$ and $r^{\mathcal{L}}_S : \mathcal{X} \to [0, \infty) \cup \{-1\}$ such that for all $x, z \in \mathcal{X}$ if $r^{\mathcal{L}}_S(z) = \eta > 0$ and $z \in \mathbf{B}^o_{\mathcal{M}}(x, \eta)$ then $\ell^{h^*}(h^{\mathcal{L}}_S, x, z) = 0$. Further, if $r^{\mathcal{L}}_S(z) = 0$, then $h^*(z) = h^{\mathcal{L}}_S(z)$.*

Note that $\mathcal{L}$ outputs a prediction and a real value $r$ (the "reliability radius") for any test input. $r = -1$ corresponds to abstention (even in the absence of perturbation) i.e. when the learner is incapable of giving a reliability guarantee for that prediction), and $r = \eta > 0$ is a guarantee from the learner that if the adversary's attack is in $\mathbf{B}^o_{\mathcal{M}}(x, \eta)$ then we are correct i.e. if an adversary changes the original test point $x$ to $z$, the attack will not succeed if the adversarial budget is less than $\eta$. Lastly, when $r = 0$, the learner provides a guarantee that the learner's prediction at $z$ is correct.

**Definition 2 (Robustly-reliable region w.r.t. $\mathcal{M}$-ball attacks)** *For a robustly-reliable learner $\mathcal{L}$ w.r.t. $\mathcal{M}$-ball attacks for sample $S$, hypothesis space $\mathcal{H}$ and robust loss function $\ell$ defined above, the robustly-reliable region of $\mathcal{L}$ at a reliability level $\eta$ is defined as $RR^{\mathcal{L}}(S, \eta) = \{x \in \mathcal{X} \mid r^{\mathcal{L}}_S(x) \geq \eta\}$ for sample $S$ and $\eta \geq 0$.*

The robustly-reliable region contains all points with a reliability guarantee of at least $\eta$. A natural goal is to find a robustly-reliable learner $\mathcal{L}$ that has the largest robustly-reliable region possible. Remarkably, it is possible to design robustly-reliable learners that achieve the pointwise optimal robustly-reliable region. A key concept used in this characterization is the disagreement region and disagreement coefficient introduced in the context of active learning [Balcan et al., 2006, Hanneke, 2007, Balcan et al., 2009].

Overall, this framework develops a significant and impactful generalization of the original notion of the reliability by Rivest and Sloan [1988]. Balcan et al. [2022] show that the framework can be applied to poisoning attacks as well. Blum and Saless [2024] show how to obtain meaningful reliability guarantees for richer and more complex concept classes where the disagreement regions may be very large. Blum et al. [2025] provide an approach to make the robustly-reliable learners even more trustworthy, by providing short certificates that may be easily verified by the users of learning model to convince themselves of the correctness of their reliability guarantee.

These works lay a fertile groundwork for studying several interesting questions and directions.

## 3 Vision for future research

The above initial works open up a compelling research program for provably reliable machine learning: per-instance, certifiably correct, and interpretable predictions under adversarial uncertainty. We see several major opportunities ahead:

1. *Beyond classification.* Extending robustly-reliable learning to other learning settings including more complex settings such as structured prediction, and reinforcement learning. For example, abstaining in structured tasks could mean returning partial predictions for which the learner is reliable.

2. *Making the algorithms more scalable and implementable with meaningful guarantees in the context of modern machine learning.* Current algorithms rely on ERM or combinatorial characterizations. Bridging these guarantees with neural architectures requires new relaxations, approximate certification techniques, and integration with uncertainty estimation (e.g., conformal prediction). Blum and Saless [2024] make progress in this direction by handling more rich classes and giving guarantees where the disagreement regions may be very large.

3. *Reliability with strategic agents.* Strategic classification (e.g. [Brückner and Scheffer, 2011, Hardt et al., 2016, Kleinberg and Raghavan, 2020, Braverman and Garg, 2020, Ahmadi et al., 2021, Attias et al., 2025, Sharma and Sun, 2025]) is a growing line of work where machine learning models are applied to classify rational agents that may respond to the model in order to change their classification. Per-point reliability guarantees would be a valuable and interesting addition to this literature, and Balcan et al. [2023b] give some initial results showing this is possible.

4. *Handling more scenarios with stronger guarantees.* Extending the analysis to richer forms of shift including concept drift, stronger guarantees like subgroup fairness reliability, and less worst-case guarantees that improve with the niceness of the distribution to capture more realistic scenarios.

5. *Active and Human-in-the-Loop learning.* How can abstention and proofs be leveraged in interactive settings where humans provide feedback? Reliable active learning [Balcan et al., 2022] could reduce labeling effort while maintaining guarantees.

6. *From worst-case to average-case or typical-case.* While worst-case robustness is principled, it may be overly pessimistic. Hybrid models—smoothed analysis, stochastic adversaries, or data-driven robustness radii capturing actual adversarial behavior [Balcan et al., 2023a]—could provide a spectrum of reliability-efficiency tradeoffs.

7. *Explanatory certificates.* Developing algorithms that output short, human-checkable reasonings and explanations in more contexts could be a powerful way to design more trustworthy approaches to machine learning. Blum et al. [2025] initiate this direction and have several concrete open questions.

8. *Reliable language generation.* A recent growing line of work studies language generation in the limit [Kleinberg and Mullainathan, 2024, Charikar and Pabbaraju, 2025, Kleinberg and Wei, 2025] as well as fundamental limits of language generation [Kalavasis et al., 2025]. It is natural to ask whether one can give reliability guarantees against training-time (data poisoning) as well as test-time (e.g. jailbreaking or prompt injection) attacks.

9. *Learning verification models to allow trustable usage of existing large models.* Training large language models and large reasoning models currently involves extremely large amounts of computing resources. An alternative to re-training them with reliability guarantees is to learn sound verification models that only accept correct outputs or reasoning steps. Recent work [Balcan et al., 2025] initiates this highly relevant direction by learning verifiers with different strengths of verification guarantees for Chain-of-Thought reasoning generation [Malach, 2024, Joshi et al., 2025].

10. *Unified models.* Developing a single framework handling multiple types of adversaries and attacks, possibly with resource-bounded adversaries or probabilistic corruption models to allow for positive results on reliability in the more challenging but very relevant models. This could involve connecting poisoning and test-time perturbations with backdoor attacks [Khaddaj et al., 2023], or studying combinations of strategic (self-interested) and adversarial (seeking specific harmful goals) elements.

## 4   Conclusion

Robustly-reliable learning is emerging as a strong foundational theory for safe machine learning. The ambitious vision presented by this line of work is to build systems that not only achieve high average accuracy but also certify the correctness of each prediction, resist adversarial interference even in their strongest instance-targeted forms, adapt across distributions, and provide concise explanations. Achieving this will require new complexity measures and analytical methods, algorithmic techniques, and connections to practice. If successful, it could redefine what it means for machine learning to be trustworthy.

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

# A Robustly-reliable learning under poisoning attacks

**Setup**. Let $\mathcal{D}$ denote a data distribution over $\mathcal{X} \times \mathcal{Y}$, where $\mathcal{X}$ is the instance space and $\mathcal{Y} = \{0, 1\}$ is the label space. Let $\mathcal{H} \subset \mathcal{Y}^{\mathcal{X}}$ be the concept space. Assume the realizable case, that is, for some $h^* \in \mathcal{H}$, for any $(x, y)$ in the support $\mathrm{supp}(\mathcal{D})$ of the data distribution $\mathcal{D}$, we have $y = h^*(x)$ (Balcan et al. [2022] also study the non-realizable case). The learner $\mathcal{L}$ has access to a corruption $S'$ of a sample $S \sim \mathcal{D}^m$, and is expected to output a hypothesis $h_{\mathcal{L}(S')} \in \mathcal{Y}^{\mathcal{X}}$ (*proper* with respect to $\mathcal{H}$ if $h_{\mathcal{L}(S')} \in \mathcal{H}$). We will use the 0-1 loss, i.e. $\ell(h, (x, y)) = \mathbb{I}[h(x) \neq y]$. For a fixed (possibly corrupted) sample $S'$, let $\mathrm{err}_{S'}(h)$ denote the average empirical loss for hypothesis $h$, i.e. $\mathrm{err}_{S'}(h) = \frac{1}{|S'|} \sum_{(x,y) \in S'} \ell(h, (x, y))$. Similarly define $\mathrm{err}_{\mathcal{D}}(h) = \mathbb{E}_{(x,y)\sim\mathcal{D}}[\ell(h, (x, h^*(x)))]$. We will consider a class of attacks where the adversary can make arbitrary corruptions to up to an $\eta$ fraction of the training sample $S$. We formalize the adversary below.

**Adversary**. Let $d(S, S') = 1 - \frac{|S \cap S'|}{m} \in [0, 1]$ denote the normalized Hamming distance between two samples $S, S'$ with $m = |S| = |S'|$. Let $A(S)$ denote the sample corrupted by adversary $A$. For $\eta \in [0, 1]$, let $\mathcal{A}_\eta$ be the set of adversaries with corruption budget $\eta$ and $\mathcal{A}_\eta(S) = \{S' \mid d(S, S') \leq \eta\}$ denotes the possible corrupted training samples under an attack from an adversary in $\mathcal{A}_\eta$. Intuitively, if the given sample is $S'$, we would like to give guarantees for learning when $S' \in \mathcal{A}_\eta(S)$ for some (realizable) uncorrupted sample $S$. Also we will use the convention that $\mathcal{A}_\eta(S) = \{\}$ for $\eta < 0$ to allow the learner to sometimes predict without any guarantees (cf. Definition 3). Note that the adversary can change both $x$ and $y$ in an example $(x, y)$ it chooses to corrupt, and can arbitrarily select which $\eta$ fraction to corrupt as in Bshouty et al. [2002].

We now state the definition of a *robustly-reliable* learner in the presence of instance-targeted attacks. For any given test example $x$, this learner outputs both a prediction $y$ and a "reliability level" $\eta_x$, such that $y$ is guaranteed to be correct so long as $h^* \in \mathcal{H}$ and the adversary's corruption budget is $\leq \eta_x$. This learner then only gets credit for predictions for which it provides a reliability guarantee of at least a desired value $\eta$.

**Definition 3 (Robustly-reliable learner)** *A learner $\mathcal{L}$ is **robustly-reliable** for sample $S'$ w.r.t. concept space $\mathcal{H}$ if, given $S'$, the learner outputs a function $\mathcal{L}_{S'} : \mathcal{X} \to \mathcal{Y} \times \mathbb{R}$ such that for all $x \in \mathcal{X}$ if $\mathcal{L}_{S'}(x) = (y, \eta)$ and if $S' \in \mathcal{A}_\eta(S)$ for some sample $S$ labeled by concept $h^* \in \mathcal{H}$, then $y = h^*(x)$. Note that if $\eta < 0$, then $\mathcal{A}_\eta(S) = \{\}$ and the above condition imposes no requirement on the learner's prediction. If $\mathcal{L}_{S'}(x) = (y, \eta)$ then let $h_{\mathcal{L}(S')}(x) = y$.*

*Given sample $S$ labeled by $h^*$, the $\eta$-**robustly-reliable region** $RR^{\mathcal{L}}(S, h^*, \eta)$ for learner $\mathcal{L}$ is the set of points $x \in \mathcal{X}$ for which given any $S' \in \mathcal{A}_\eta(S)$ we have that $\mathcal{L}_{S'}(x) = (y, \eta')$ with $\eta' \geq \eta$. More generally, for a class of adversaries $\mathcal{A}$ with budget $\eta$, $RR_{\mathcal{A}}^{\mathcal{L}}(S, h^*, \eta)$ is the set of points $x \in \mathcal{X}$ for which given any $S' \in \mathcal{A}(S)$ we have that $\mathcal{L}_{S'}(x) = (y, \eta')$ with $\eta' \geq \eta$. We also define the* **empirical $\eta$-robustly-reliable region** $\widehat{RR}^{\mathcal{L}}(S', \eta) = \{x \in \mathcal{X} : \mathcal{L}_{S'}(x) = (y, \eta')$ *for some* $\eta' \geq \eta\}$. *So,* $RR_{\mathcal{A}}^{\mathcal{L}}(S, h^*, \eta) = \cap_{S' \in \mathcal{A}(S)} \widehat{RR}^{\mathcal{L}}(S', \eta)$.

The requirement of security to targeted attacks appears both in the definition of a robustly-reliable learner as well as in its measure of performance, the $\eta$-robustly-reliable region. First, if a robustly-reliable learner outputs $(y, \eta)$ on input $x$, then $y$ must be correct even if an $\eta$ fraction of the training data had been corrupted specifically to target $x$ (the corruptions can be different for different inputs $x$). Second, for a point $x$ to be in the $\eta$-robustly-reliable region, it must be the case that for any $\eta$-corrupted training set $S' \in \mathcal{A}_\eta(S)$ (even if $S'$ is a targeted attack on $x$) we have $\mathcal{L}_{S'}(x) = (y, \eta')$ for some $\eta' \geq \eta$. So, points in the $\eta$-robustly-reliable region are points an adversary cannot successfully target with a budget of $\eta$ or less.

