# OpenReview forum: "Learning reliably under adversarial attacks, distribution shifts and strategic agents"
_NeurIPS.cc/2025/Workshop/Reliable_ML — NeurIPS 2025 - Reliable ML Workshop_

### Official Review · Reviewer_RnfJ · 2025-09-18
**Summary of robustly-reliable learning, future directions are too vague/broad**

**Rating:** 5
**Confidence:** 3

**Review:**

Summary: The paper summarizes current findings and bounds within the field of robustly-reliable learning and proposes a forward-thinking agenda on future directions in this field. The emphasis is on not just ensuring average reliability but ensuring per-prediction reliability.

Strengths: This paper is very relevant to this workshop and does provide a concise and fairly thorough and theoretical summary of robustly-reliable learning.

Weaknesses / Limitations: The vision for future research feels a bit too vague and unspecific. The intro and motivation focus mainly on LLMs as the models that need to be more robust/reliable, but the future directions are scattered across all parts of ML and to me, feels a bit too broad. I would rather see 3-4 directions and elaborate more on each of those.

---

### Official Review · Reviewer_te4L · 2025-09-19
**Summary and some feedbacks**

**Rating:** 7
**Confidence:** 3

**Review:**

This paper presents a survey and forward-looking perspective on the emerging framework of robustly-reliable learning. Unlike traditional PAC guarantees or adversarial robustness frameworks that mainly address average error, robustly-reliable learning emphasizes per-instance guarantees of correctness, often accompanied by abstention mechanisms and reliability certificates.

The paper reviews prior work on reliable learning (Rivest \& Sloan, 1988; Chow’s reject option), selective classification, and recent extensions that provide guarantees under poisoning attacks, test-time adversaries, distribution shifts, and strategic agents. It highlights key results such as the characterization of robustly reliable learners in terms of the disagreement coefficient and the development of algorithms that achieve optimal reliability guarantees in various settings.
The paper positions robustly reliable learning as a unifying theoretical foundation for trustworthy ML in adversarial and uncertain environments.
As a survey paper, it effectively synthesizes a range of results and perspectives on reliable learning. However, this paper is primarily a perspective/survey rather than presenting new theorems, algorithms, or empirical validation. Its contribution is more conceptual synthesis than technical advancement.